# AdaFlow: Efficient Long Video Editing via Adaptive Attention Slimming And Keyframe Selection

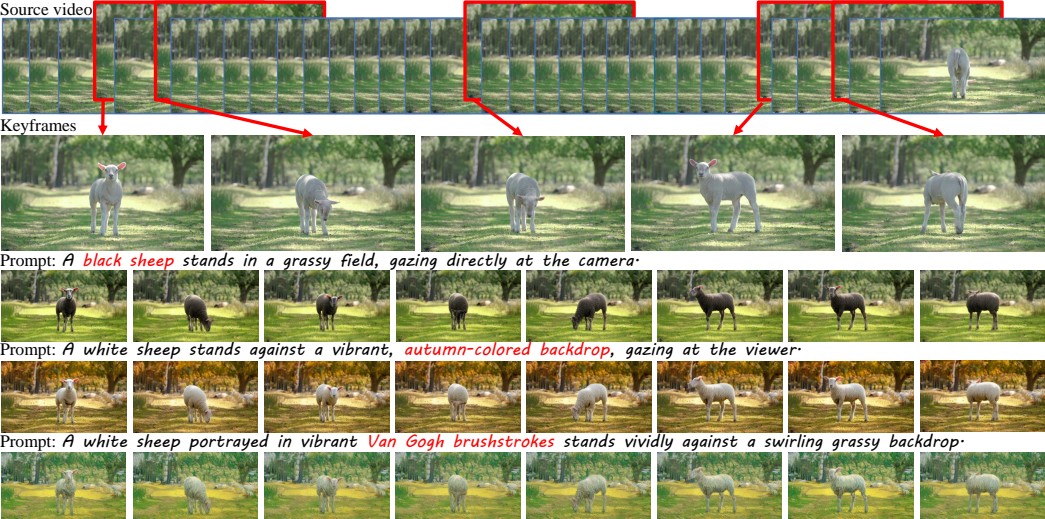

Figure 1: The proposed AdaFlow can support the text-driven video editing of more than $1k$ frames in one inference, which can be the change of the primary subjects, background, or overall style of the video. Meanwhile, AdaFlow can also adaptively select the representative frames in different video clips for keyframe translation, ensuring the continuity and quality of long video editing.

## Abstract

Text-driven video editing is an emerging research hot spot in deep learning. Despite great progress, long video editing is still notoriously challenging mainly due to excessive memory overhead. To tackle this problem, recent efforts have simplified this task into a two-step process of keyframe translation and interpolation generation, enabling the editing of more frames. However, the token-wise keyframe translation still plagues the upper limit of video length. In this paper, we propose a novel and training-free approach towards efficient and effective long video editing, termed *AdaFlow*. We first reveal that not all tokens of video frames hold equal importance for keyframe-consistency editing, based on which we propose an *Adaptive Attention Slimming* scheme for AdaFlow to squeeze the $KV$ sequence of extended self-attention. This enhancement allows AdaFlow to increase the number of keyframes for translations by an order of magnitude. In addition, an Adaptive Keyframe Selection scheme is also equipped to select the representative frames for joint editing, further improving generation quality. With these innovative designs, AdaFlow achieves high-quality long video editing of minutes in one inference, *i.e.*, more than $1k$ frames on one A800 GPU, which is about ten times longer than the compared methods. To validate AdaFlow, we also build a new benchmark for long video editing with high-quality annotations, termed *LongV-EVAL*. The experimental results show that our AdaFlow can achieve obvious advantages in both the efficiency and quality of long video editing. Our code is anonymously released at https://anonymous.4open.science/r/AdaFlow-C28F.

# 1 INTRODUCTION

Recent years have witnessed the great success of diffusion-based models in high-quality text-driven image generation and editing (Ho et al., 2020; Hertz et al., 2022; Couairon et al., 2022; Tumanyan et al., 2023; Brooks et al., 2023; Tewel et al., 2024). More recently, the rapid development of image diffusion models also sparks an influx of attention to text-driven video editing (Geyer et al., 2023; Cong et al., 2023; Qi et al., 2023). As a milestone in the research of *AI Generated Content* (AIGC), text-driven video editing can well broaden the application scope of diffusion models, such as *animation creation*, *virtual try-on*, and *video effects enhancement*. However, compared with the well-studied image editing, text-driven video editing is still far from satisfactory due to its high requirement of frame-wise consistency (Wu et al., 2023b; Qi et al., 2023; Yang et al., 2023; 2024). Meanwhile, its extremely high demand for computation resources also greatly hinders development (Cong et al., 2023; Wu et al., 2023b; Kara et al., 2024).

Most existing methods (Cong et al., 2023; Wu et al., 2023b; Kara et al., 2024; Liu et al., 2024) can only support video editing of a few seconds, and long video editing is still notoriously challenging. In particular, current research often resorts to the well-trained image diffusion models for video editing via test-time tuning (Wu et al., 2023b; Liu et al., 2024) or training-free paradigms (Ceylan et al., 2023; Cong et al., 2023; Kara et al., 2024). To maintain the smoothness and consistency of edited videos, these methods primarily extend the self-attention module in diffusion models to all video frames, commonly referred to as *extended self-attention* (Geyer et al., 2023; Wu et al., 2023b). Despite its effectiveness, this solution will lead to a quadratic increase in computation as the number of video frames grows, and the token-based representations of these video frames further greatly exacerbate the memory footprint. For instance, the editing of ten video frames needs to compute extended self-attention on up to $40k$ visual tokens in the diffusion model (Geyer et al., 2023). As a result, processing only a few video frames will require a prohibitive GPU memory footprint, making existing approaches can only conduct video editing of several seconds.

To alleviate this issue, recent endeavors focus on factorizing video editing into a two-step generation task (Geyer et al., 2023; Yang et al., 2023; 2024). The first step is *keyframe translation*, which samples the video keyframes to perform extended self-attention. Afterwards, all frames are fed to the diffusion model for editing based on the translated keyframe information, often termed *interpolation generation* (Geyer et al., 2023). Compared to the direct editing on all video frames, this two-step solution only needs to perform the quadratic computation of extended self-attention for the keyframes, thus improving the number of overall editing frames from a dozen to nearly one hundred frames (Geyer et al., 2023). However, the basic mechanism of extended self-attention is still left unexplored, making these approaches (Geyer et al., 2023; Yang et al., 2023; 2024) still hard to achieve minute-long video editing in one inference. Moreover, the naive uniform sampling of keyframes (Geyer et al., 2023) also does not consider the change of video content, *e.g.*, the motion of objects or the transitions of the scene, and a large sampling interval will inevitably undermine video quality.

In this paper, we propose a novel and training-free method called *AdaFlow* for high-quality long video editing. In particular, we first observe that during extended self-attention, not all visual tokens of a video frame are equally important for maintaining frame consistency and video continuity. Only the tokens of the frame correspond to the *query* matter. In this case, *Adaptive Attention Slimming* is proposed to squeeze the less important ones in the $KV$ sequence of extended self-attention, thereby greatly alleviating the computation burden. Meanwhile, AdaFlow also introduces an *Adaptive Keyframe Selection* to pick up the frames that can well represent the edited video content, thus avoiding the translation of redundant keyframes and improving the utilization of computation resources. With these innovative designs, AdaFlow can improve the number of video frames edited by an order of magnitude, realizing true long video editing.

To well validate the proposed AdaFlow, we also propose a new long video editing benchmark to complement the existing evaluation system, termed *LongV-EVAL*. This benchmark consists of 75 videos, and they are about one minute long and cover various scenes, such as *humans*, *landscapes*, *indoor settings* and *animals*. For LongV-EVAL, we meticulously design a data annotation process, which applies multimodal large language models (Achiam et al., 2023; Lin et al., 2023) to generate three high-quality editing prompts for each video. These prompts focus on different aspects of the video, such as *primary subjects*, *background*, *overall style*, and *so on*. In terms of evaluation metrics, we follow (Sun et al., 2024) to evaluate the edited videos from the aspects of *frame quality*, *video quality*, *object consistency*, and *semantic consistency* on LongV-EVAL.

To validate AdaFlow, we conduct extensive experiments on the proposed LongV-EVAL benchmark, and also compare AdaFlow with a set of advanced video editing methods (Yang et al., 2023; Geyer et al., 2023; Cong et al., 2023; Yang et al., 2024; Kara et al., 2024). Both qualitative and quantitative results show that our AdaFlow has obvious advantages over the compared methods in terms of the efficiency and quality of long video editing. More importantly, AdaFlow can effectively conduct various high-quality edits for videos of more than 1000 frames on a single GPU, *e.g.*, changing the main object, background or overall style.

Conclusively, the contribution of this paper is threefold:

- We propose a novel and training-free video editing method called *AdaFlow* with two innovative designs, namely *Adaptive Attention Slimming* and *Adaptive Keyframe Selection*.
- The proposed AdaFlow is capable of long video editing of more than 1000 frames in one inference on a single GPU, and it also supports various editing tasks, such as the change to the background, foreground, overall styles, and so on.
- We also build a high-quality benchmark to complement the lack of long video editing evaluation, termed *LongV-EVAL*. On this benchmark, our AdaFlow shows obvious advantages over the compared methods in terms of efficiency and quality.

## 2 RELATED WORKS

### 2.1 DIFFUSION-BASED IMAGE AND VIDEO GENERATION

Diffusion models have gained significant traction in image and video generation (Rombach et al., 2022; Croitoru et al., 2023; Guo et al., 2023; Blattmann et al., 2023; Wang et al., 2024; Peng et al., 2024). In image generation, DDPM (Ho et al., 2020) and its variants (Song et al., 2020; Dhariwal & Nichol, 2021; Nichol & Dhariwal, 2021; Rombach et al., 2022; Croitoru et al., 2023; Guo et al., 2023) have demonstrated impressive results in producing detailed and realistic images. They iteratively refine noisy images, progressively improving quality and coherence.

In addition, recent advances (Ho et al., 2022a;b; Wu et al., 2023b; Blattmann et al., 2023; Wang et al., 2024) have extended diffusion models to video generation, where temporal consistency is crucial. These methods build upon the success of image-based diffusion models by incorporating temporal attention mechanisms to ensure consistency across frames. However, challenges persist, particularly with long video generation, due to the computational and memory demands of processing hundreds or thousands of frames. To address this, some methods adopt a divide-and-conquer approach, while others adopt a temporal autoregressive approach (Li et al., 2024).

### 2.2 TEXT-DRIVEN VIDEO EDITING

With the success of image and video generation, an increasing number of works have applied pretrained text-to-image diffusion models to video editing (Wang et al., 2023; Wu et al., 2023b; Ma et al., 2024; Liu et al., 2024), with the primary challenge being maintaining temporal consistency across frames. Zero-shot video editing methods have gained attention for addressing this issue. FateZero (Qi et al., 2023) introduced an attention blending module, combining attention maps from the source and edited videos during the denoising process to improve consistency. TokenFlow (Geyer et al., 2023) computes frame feature correspondences via nearest neighbors, which is similar to optical flow, enhancing coherence. Similarly, Flatten (Cong et al., 2023) proposed flow-guided attention that uses optical flow to guide attention for smoother editing. Video-P2P (Liu et al., 2024) adapted classic image editing methods to video, but editing even an 8-frame video takes over ten minutes, making it impractical for real-world applications.

Although these methods offer effective solutions for video editing, they struggle with long videos having thousands of frames. InsV2V (Cheng et al., 2023) directly trains a video-to-video model and proposes a method for long video editing, but it only edits about 20-30 frames ($\sim 1s$) at a time and stitches them together, resulting in cumulative errors and quality decline after several iterations.

In addition to processing long videos, great content modification is also a main obstacle of video editing (Cong et al., 2023; Geyer et al., 2023), such as structural modifications or adding new objects. Most motion-flow-based methods (Cong et al., 2023; Geyer et al., 2023) as well as our AdaFlow are

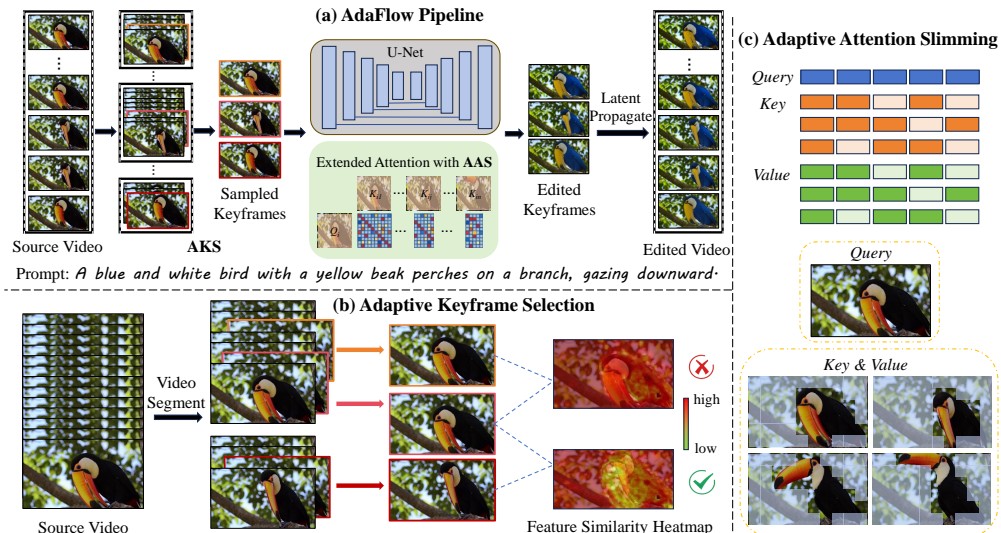

Figure 2: The framework of the proposed AdaFlow. (a) The pipeline of AdaFlow for long video editing. Given a source video and the text editing prompt, AdaFlow first applies *Adaptive Keyframe Selection* (AKS) to adaptively divide the video into clips according to its content and then sample frames for keyframe translation. Afterwards, *Adaptive Attention Slimming* (AAS) is applied to reduce the redundant tokens in *Extended Self-Attention* for keyframe translation, thereby increasing the number of frames edited. Finally, the editing information of the keyframes is propagated throughout the entire video. (b) Adaptive Keyframe Selection (AKS) truncates video clips according to the frame-wise DIFT similarities and selects the adaptive keyframes according to video clips. (c) Adaptive Attention Slimming removes the redundant tokens of frames in the $K, V$ sequence for Extended Self-attention, thereby greatly saving the GPU memory footprint for keyframe translation.

limited to this target under the training-free setting. In particular, this challenge often requires large-scale training or test-time tuning (Wu et al., 2023b; Qi et al., 2023; Gu et al., 2024), such as FateZero (Qi et al., 2023) that performs significant structural editing with test-time tuning, which is orthogonal to the contribution of this paper.

## 3 PRELIMINARY

**Diffusion Models.** *Denoising diffusion probabilistic model* (DDPM) (Ho et al., 2020) is a generative network that aims at reconstructing a forward Markov chain $\{x_1, \ldots, x_T\}$. For a data distribution $x_0 \sim q(x_0)$, the Markov transition $q(x_t|x_{t-1})$ follows a Gaussian distribution with a variance schedule $\beta_t \in (0, 1)$:

$$q\left(\boldsymbol{x}_t \mid \boldsymbol{x}_{t-1}\right) = \mathcal{N}\left(\boldsymbol{x}_t; \sqrt{1 - \beta_t}\boldsymbol{x}_{t-1}, \beta_t\mathbf{I}\right). \tag{1}$$

To generate the Markov chain $\{x_0, \ldots, x_T\}$, DDPM employs a reverse mechanism with an initial distribution $p(x_T) = \mathcal{N}(x_T; 0, I)$ and Gaussian transitions. A neural network $\epsilon_\theta$ is trained to estimate the noise, ensuring that the reverse mechanism approximates the forward process:

$$p_\theta\left(\boldsymbol{x}_{t-1} \mid \boldsymbol{x}_t\right) = \mathcal{N}\left(\boldsymbol{x}_{t-1}; \mu_\theta\left(\boldsymbol{x}_t, \boldsymbol{\tau}, t\right), \Sigma_\theta\left(\boldsymbol{x}_t, \boldsymbol{\tau}, t\right)\right), \tag{2}$$

where $\tau$ denotes the text prompt. The parameters $\mu_\theta$ and $\Sigma_\theta$ are inferred by the denoising model $\epsilon_\theta$. Latent diffusion (Rombach et al., 2022) alleviates the computational demands by executing these processes within the latent space of a *variational autoencoder* (Kingma, 2013).

**Diffusion Features.** Diffusion Features (DIFT) can extract the correspondence of images from the diffusion network $\epsilon_\theta$ without explicit supervision (Tang et al., 2023). Starting from noise $z$, a series of images $x_t$ are generated by gradual denoising through a reverse diffusion process. At each timestep $t$, the output of each layer of $\epsilon_\theta$ can be used as a feature. Larger $t$ and earlier network layers produce more semantically aware features, while smaller $t$ and later layers focus more on

low-level details. To extract DIFT from an existing image, Tang et al. (2023) propose adding noise of timestep $t$ to the real image, then inputting it into the network $\epsilon_\theta$ along with $t$ to extract the latent of the intermediate layer as DIFT. This method predicts corresponding points between two images, and can even generate correct correspondences across different domains.

**Extended Self-Attention.** To ensure video smoothness and coherence, the self-attention block of an image diffusion model must edit all frames simultaneously (Wu et al., 2023b; Geyer et al., 2023). In this case, *Extended Self-Attention* (ESA) is introduced to maintain the coherence and temporal consistency of the video. For the latent of the $i$-th frame at timestep $t$, denoted as $z_t^i$, the attention score is computed between the $i$-th frame and all other $n$ frames. Mathematically, the extended self-attention can be formulated as

$$\text{Attention}(Q_i, K_{1:n}, V_{1:n}) = \text{Softmax}\left(\frac{Q_i K_{1:n}^T}{\sqrt{d}}\right) \cdot V_{1:n}, \tag{3}$$

where $Q_i = W^Q z_t^i, K_{1:n} = W^K z_t^{1:n}, V_{1:n} = W^V z_t^{1:n}$. Here, $W^Q, W^K$, and $W^V$ are the weighted matrices identical to those used in the self-attention layers of the image diffusion model.

## 4 METHOD

Given a source video of $n$ frames, $\mathcal{I} = [\boldsymbol{I}_1, ..., \boldsymbol{I}_n], \boldsymbol{I}_i \in \mathbb{R}^{H \times W}$, where $H \times W$ denotes the resolution, and a text prompt $\mathcal{P}$ describing the editing task, we first use a pre-trained text-to-image diffusion model $\epsilon_\theta$ to extract its diffusion features, denoted as $\mathcal{F} = [\boldsymbol{F}_1, ..., \boldsymbol{F}_n], \boldsymbol{F}_i \in \mathbb{R}^{h \times w \times d}$. Based on the obtained diffusion features $\mathcal{F}$, AdaFlow employs *Adaptive Keyframe Selection* (Sec.4.1) to divide the video into multiple clips based on the content. For each clip that consists of consecutive frames with similar content, one frame is then sampled as a keyframe at each timestep, and all keyframes are edited simultaneously using $\epsilon_\theta$. To edit videos as long as possible, AdaFlow then applies *Adaptive Attention Slimming* to reduce the length of $KV$ sequences in extended self-attention for keyframe translation (Sec. 4.2). Finally, the information from translated keyframes is propagated to the remaining frames to ensure smoothness and continuity throughout the edited video, which is denoted as $\mathcal{J} = [\boldsymbol{J}^1, ..., \boldsymbol{J}^n]$ (Sec. 4.3).

**Pre-processing.** Given the source video $\mathcal{I}$, we first use a pre-trained text-to-image diffusion model $\epsilon_\theta$ to extract the diffusion features of each frame $\boldsymbol{I}_i$, resulting in $\mathcal{F} = [\boldsymbol{F}_1, ..., \boldsymbol{F}_n]$. Afterwards, we use the diffusion model $\epsilon_\theta$ to perform DDIM inversion (Song et al., 2020) on each frame $\boldsymbol{I}_i$ to obtain a sequence of latents, which will be used in the subsequent editing.

### 4.1 ADAPTIVE KEYFRAME SELECTION

Keyframe selection is critical for long video editing, which however is often ignored in previous works (Wu et al., 2023b; Cong et al., 2023; Liu et al., 2024). When the visual content of a given video changes rapidly, keyframe samplings at shorter intervals are usually required to ensure the editing quality (Geyer et al., 2023), but it will result in a large number of redundant frames for editing. To address this issue, we propose *Adaptive Keyframe Selection* (AKS) based on the video content. In particular, consecutive and similar frames are grouped into clips allowing for more informed keyframe sampling. In periods where the visual content changes rapidly, keyframes can be selected more densely, whereas fewer frames are required for clips with less dynamic content. In this case, AKS can retain editing quality while reducing the computational burden, particularly for videos with little variation.

In practice, Adaptive Keyframe Selection (AKS) resorts to DIFT features for frame-wise similarity. DIFT can effectively match corresponding points between images (Tang et al., 2023). It is shown that when two images are not very similar, the confidence level of the matching decreases significantly. Based on this principle, AKS uses DIFT to quickly assess the degree of change in a video. As shown in Fig.2 (b), we can obtain a heatmap to represent the temporal dynamics (Brooks et al., 2022) between frames using DIFT. When there is a noticeable shift in the angle of objects in the frame or a sudden appearance of new objects, these regions will show brighter colors in the heatmap.

Concretely, to compute the heatmap $H_{i,j} \in \mathbb{R}^{h \times w}$ of the temporal dynamics between the $i$-th frame and the $j$-th frame, we compute the token-wise cosine similarity using their DIFT features. For a token $p$ in the $i$-th frame and a token $q$ in the $j$-th frame, whose feature vectors are $f_i^p \in \boldsymbol{F}_i$ and

$f_j^q \in \mathbf{F}_j$, the cosine similarity $CS(\cdot)$ is computed by

$$CS(f_i^p, f_j^q) = \frac{f_i^p \cdot f_j^q}{\|f_i^p\|\|f_j^q\|}. \tag{4}$$

Then the token $q^*$ most similar to the token $p$ is obtained by

$$q^* = \arg\max_{q \in \boldsymbol{T}_j} CS(f_i^p, f_j^q), \tag{5}$$

where $\boldsymbol{T}_j$ denotes all tokens corresponding to the $j$-th frame.

Finally, the value corresponding to token $p$ in the heatmap is

$$H_{i,j}^p = CS(f_i^p, f_j^{q^*}). \tag{6}$$

After obtaining the heatmaps of a video, we can use them to segment clips that consist of consecutive frames with similar content, of which procedure is described in Algorithm 1. In principle, we determine the partition points of the video by calculating the similarity between video frames. Specifically, we traverse the sequence of video frames and calculate the similarity heatmap for the frame pair. If the mean value of the heatmap between a pair of frames is smaller than a defined threshold, or if the sliding window finds the mean value below the threshold at any point, the current frame will be marked as the start of a new clip. Then, we continue traversing from the next possible starting point until the entire video is processed. Finally, we obtain the starting indices of all clips $\mathcal{S} = \{\boldsymbol{s}_1, ..., \boldsymbol{s}_M\}$, where $M$ represents the total number of clips.

In Appendix E, we visualize the content-aware video partitioning with a $y - t$ plot. As shown in Fig.7, the adaptively partitioned video clips are similar within each part, but the partitioning points are accurately positioned where the video content undergoes rapid changes.

---

**Algorithm 1** Adaptive Video Partitioning

**Require:**
    $\mathcal{F}$: DIFT for each frame,
    $n$: Number of frames,
    $l$: Sliding window size,
    $s$: Step size,
    $ms$: Mean threshold,
    $ws$: Window threshold.
1: segments = [ ]
2: $i = 1, j = 2$
3: **while** $j < n$ **do**
4:     calculate $H_{i,j}$ with $F_i, F_j$
5:     **if** mean$(H_{i,j}) < ms$
       **or** not window_check$(H_{i,j}, l, s, ws)$ **then**
6:        segments.append($i$)
7:        $i = j + 1$
8:        $j = i + 1$
9:     **else**
10:        $j = j + 1$
11:     **end if**
12: **end while**
13: **return** segments

---

After partitioning, we can directly select a frame from each partition at each timestep, obtaining a total of $M$ keyframes, denoted as $\mathcal{K} = [\boldsymbol{I}_{k_1}, ..., \boldsymbol{I}_{k_M}]$, which satisfies $\boldsymbol{s}_i \leq k_i < \boldsymbol{s}_{i+1}$.

## 4.2 ADAPTIVE ATTENTION SLIMMING

As mentioned in Section 3, we use extended self-attention for keyframe translation, thereby ensuring the smoothness and continuity of edited videos. However, extended self-attention involves the concatenation of $KV$ tokens of all frames, resulting in a quadratic increase in computation. Moreover, the extremely high GPU memory footprint becomes a bottleneck for long video editing. Besides, if the number of keyframes is severely limited, it will significantly hinder the length of the editable video and adversely affect the editing quality. To address this issue, we propose a novel *Adaptive Attention Slimming* (AAS) method to reduce the $KV$ sequence of extended self-attention, which can significantly improve computational efficiency without affecting video editing quality.

Concretely, given one keyframe $I_{k_i}$, similar to Eq.6, we use DIFT to calculate $M$ cosine similarity heatmaps between this keyframe and all other keyframes, denoted as $H = \{H_{k_1, k_i}, H_{k_2, k_i}, \ldots, H_{k_M, k_i}\}$. From these heatmaps, we select the $m$ pixel positions with the highest values. For $K$ and $V$ in extended self-attention, we retain only the tokens corresponding to these $m$ positions and obtain new $\widetilde{K}_{k_1:k_M}$ and $\widetilde{V}_{k_1:k_M}$, of which length is much shorter than the default ones. Afterwards, the slimmed Extended Self-attention is defined by

$$\text{Attention}(Q_i, \widetilde{K}_{k_1:k_M}, \widetilde{V}_{k_1:k_M}) = \text{Softmax}\left(\frac{Q_i \widetilde{K}_{k_1:k_M}^T}{\sqrt{d}}\right) \cdot \widetilde{V}_{k_1:k_M}. \tag{7}$$

For ease of subsequent calculations, we abbreviate $\text{Attention}(Q_i, \widetilde{K}_{k_1:k_M}, \widetilde{V}_{k_1:k_M})$ as $\mathcal{A}_i$.

In Appendix D, we visualize the relationship between the retained tokens in the *key/value* pairs and the *query*. It can be intuitively observed that the KV tokens more related to the *query* frames are retained more, while the ones different from the *query* are often discarded. It is because over longer time spans, more content becomes dissimilar to the *query*, and attending to these contents does not significantly improve the generation quality and consistency of the *query* frames. Conversely, frames closer to the *query* are crucial for maintaining the video's coherence. Therefore, the proposed AAS can save computational resources and minimize the impact on video editing quality.

### 4.3 FEATURE-MATCHED LATENT PROPAGATION

Similar to TokenFlow (Geyer et al., 2023), we propagate the generation of keyframes to non-keyframes based on the token correspondences obtained from the source video, thus generating a continuous and smooth video. However, unlike TokenFlow (Geyer et al., 2023), which requires the calculations of token correspondences at each timestep and every self-attention operation, our method only needs to compute the correspondences once before editing, and saves them for the use in following timesteps. This setting greatly simplifies the computational process.

Specifically, given the source video and the obtained video clips, we compute token correspondences between every two frames within the same clip. The formula for calculating the spatial position $p$ of the $i$-th frame corresponding to the $j$-th frame is the same as Eq.5. For convenience, we express the correspondence between the position $p$ in the $i$-th frame and the position $q^*$ in the $j$-th frame as

$$\phi_{ij}(p) = q^*. \tag{8}$$

For each non-keyframe $i$, there is a keyframe $j$ within the same video clip. Through the calculation above, we can map each token in $\mathcal{A}_i$ to a corresponding token in $\mathcal{A}_j$, which can be expressed as

$$\mathcal{A}_i[p] = \mathcal{A}_j[\phi_{ij}(p)]. \tag{9}$$

For cases where there may be an inconsistent size between $F_i$ and the output latent of self-attention $\mathcal{A}_i$, a simple resize operation is sufficient and will not affect the generation quality.

Note that, due to the principle of motion-flow-based video editing (Geyer et al., 2023; Cong et al., 2023), our AdaFlow is still limited to the significant editing of video content, such as structural modifications or adding new objects.

## 5 EXPERIMENTS

### 5.1 LONG VIDEO EDITING EVALUATION BENCHMARK

In this paper, we also propose a new long video editing benchmark considering the lack of specific evaluation of text-driven long video editing, termed ***LongV-EVAL***. Concretely, we collected 75 videos of approximately 1 minute in length, which cover various domains such as landscapes, people, and animals. We then annotate the videos using Video-LLaVA (Lin et al., 2023) and GPT-4 (Achiam et al., 2023), generating three high-quality video editing prompts for each video. These three prompts focus on different aspects of editing, *i.e.*, the change to foreground, background or overall style. More details of this benchmark are described in Appendix A.

In terms of evaluation, we follow Sun et al. (2024) to use four quantitative evaluation metrics: (1) **Frames Quality** (FQ): Before considering all video frames together, the quality of each individual frame forms the foundation for determining the overall video quality. We use the LAION aesthetic predictor (Schuhmann et al., 2021), which is aligned with human rankings, for image-level quality assessment. This predictor estimates aspects such as composition, richness, artistry, and visual appeal of the images. We take the average aesthetic score of all frames as the overall quality score of the video. (2) **Video Quality** (VQ): We use the DOVER score (Wu et al., 2023a) for video-level quality assessment. DOVER is the most advanced video evaluation method trained on a large-scale human-ranked video dataset. It can evaluate aspects such as artifacts, distortions, blurriness, and incoherence. (3) **Object Consistency** (OC): In addition to evaluating overall video quality, maintaining object consistency in long video editing is also important. We use DINO (Caron et al., 2021), a self-supervised pre-trained image embedding model, to calculate frame-to-frame similarity at the

Table 1: Comparisons between AdaFlow and the state-of-the-art methods on LongV-EVAL. Here, *Mins/Video* denotes the average number of minutes of video editing. *FQ*, *VQ*, *OC*, and *SC* denote *frame quality*, *video quality*, *object consistency*, and *semantic consistency*, respectively.

| Method | FQ↑ | VQ↑ | OC↑ | SC↑ | Mins/Video↓ |
|---|---|---|---|---|---|
| Rerender(Yang et al., 2023) | 5.36 | 0.638 | 0.942 | 0.961 | 52 |
| TokenFlow(Geyer et al., 2023) | 5.30 | 0.808 | 0.947 | 0.966 | 40 |
| FLATTEN(Cong et al., 2023) | 5.05 | 0.637 | 0.882 | 0.931 | 80 |
| RAVE(Kara et al., 2024) | 5.17 | 0.677 | 0.861 | 0.909 | 83 |
| FRESCO(Yang et al., 2024) | **5.65** | 0.820 | 0.930 | 0.954 | 47 |
| AdaFlow (ours) | 5.43 | **0.839** | **0.953** | **0.969** | **24** |

Table 2: User study. 18 participants are asked to evaluate the edited videos of different methods in terms of video quality and temporal consistency. The values are the percentages of choices.

| Metrics | Rerender | TokenFlow | FLATTEN | FRESCO | RAVE | AdaFlow (Ours) |
|---|---|---|---|---|---|---|
| **Video Quality** | 0.0% | 12.5% | 1.8% | 4.5% | 3.6% | **77.7%** |
| **Temporal Consistency** | 0.0% | 10.7% | 1.8% | 11.6% | 0.0% | **75.9%** |

object level. (4) **Semantic Consistency** (SC): CLIP (Radford et al., 2021) visual embeddings are widely used to capture the semantic information of images. The cosine similarity of CLIP embeddings between adjacent frames is a standard metric for evaluating the frame-to-frame consistency and overall smoothness of a video.

## 5.2 EXPERIMENTAL SETUPS

In our experiments, we use the official pre-trained weights of Stable Diffusion (SD) 2.1 (Rombach et al., 2022) as the text-to-image model. We employ DDIM Inversion with 50 timesteps and denoising with 50 timesteps. For image editing, we adopt PnP-Diffusion (Tumanyan et al., 2023). When extracting DIFT, we select the features corresponding to t=0 for each frame of the source video (Tang et al., 2023), which are extracted from the intermediate layer of the 2D Unet Decoder. During editing, the video resolution is set to 384x672. For keyframe selection, the average similarity threshold is set to 0.75, and the similarity threshold within the sliding window is set to 0.6. The sliding window has a side length of 42 pixels, with a step size of 21 pixels per slide. For joint editing of keyframes, if the number of keyframes exceeds 14, pruning is initiated. We consistently retain the token count corresponding to 14 frames, with the degree of pruning increasing as the number of keyframes increases. All our experiments are conducted on an NVIDIA A800 80GB GPU.

In our experiments, we mainly compare our AdaFlow with five advanced video editing methods, including Rerender (Yang et al., 2023), TokenFlow (Geyer et al., 2023), FLATTEN (Cong et al., 2023), FRESCO (Yang et al., 2024), and RAVE (Kara et al., 2024). For these baselines, we use the default settings provided in their official GitHub repositories. Since TokenFlow, FLATTEN, and RAVE are unable to edit long videos in a single inference, we segment the long videos for editing. Based on their computational resource usage, we edit 128, 32, and 16 frames at a time.

## 5.3 QUANTITATIVE ANALYSIS

In Tab.1, we first quantitatively compare the proposed AdaFlow with a set of the latest video editing methods (Yang et al., 2023; Geyer et al., 2023; Cong et al., 2023; Yang et al., 2024; Kara et al., 2024) on LongV-EVAL. In particular, we accomplish the long video editing of the compared methods in multiple inferences due to the limit of GPU memory. As can be seen, our AdaFlow achieves better performance than the compared methods in terms of video quality, object consistency, and semantic consistency. Although it is slightly inferior to FRESCO (Yang et al., 2024) in frame quality, FRESCO has a large gap between the edited video and the source video, according to the visualization of Fig.3. In addition to delivering excellent editing quality, our AdaFlow not only enables the editing of longer videos but also achieves much higher efficiency through its innovative designs. As

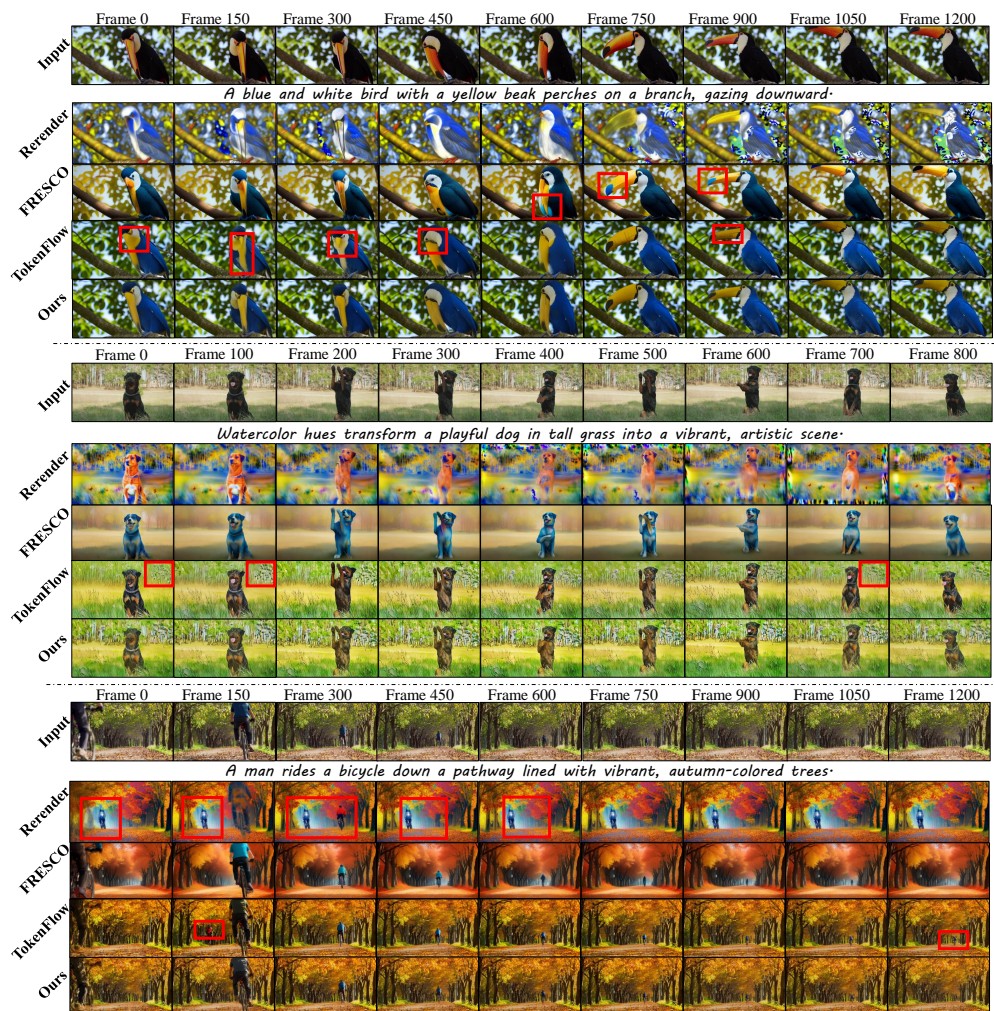

Figure 3: Comparisons of AdaFlow with a set of advanced video editing methods. The red box refers to the failed editing of the methods, *e.g.*, the changes of objects or background, or the inconsistency between frames. Compared with the other methods, our AdaFlow can not only process videos of up to $1k$ frames in one inference but also can well keep the quality and continuity of edited videos.

shown in the last column of Tab.1, our method takes an average of 24 minutes to edit a video, while the baselines take at least 40 minutes, almost twice as long as ours.

In addition to the measurable metrics of LongV-EVAL, we also conduct a comprehensive user study to compare our AdaFlow with other methods in Tab.2. In practice, we invited 18 participants to choose their preferred videos edited by different methods based on two metrics, *i.e.*, video quality and temporal consistency. We randomly selected 20 sets of video-text data for the user study. Each set contains 6 videos for comparison, so each participant needs to view 120 long videos and make 40 choices. The specific evaluation criteria are given in Appendix C. Considering the participants' attention span, we believe this is an appropriate amount of data. As shown in Tab.2, it is evident that our method is the most favored in terms of two metrics. Overall, these results well validate the efficiency and effectiveness of our AdaFlow for long video editing.

## 5.4 QUALITATIVE RESULTS

To better evaluate the effectiveness of our AdaFlow, we visualize its key steps in Fig.1 and also compare its results with a set of the latest video editing methods in Fig.3. As shown in Fig.1, for a video approximately 1000 frames long, AdaFlow adaptively segments the video clips based on con-

Figure 4: Ablation Study for Adaptive Keyframe Selection (AKS). AKS can capture the abrupt changes of edited videos to ensure the editing quality, *e.g.*, the appearance of the car (left), or the cat yawning suddenly (right). Without AKS, the rapidly changing parts of the video are often blurry.

tent, and then selects keyframes (Row 2) accurately and effectively perform text-guided keyframe translation. For instance, transforming a white sheep into a black sheep (Row 3), changing a lush green scene into an autumn atmosphere (Row 4), or translating the video into the *Van Gogh* style (Row 5). Each edit strictly follows the text prompt and maintains the consistency with the source video for the parts that do not require changing. More visualization can be found in Appendix B.

In Fig.3, we compare the edited videos by AdaFlow with those of Rerender (Yang et al., 2023), FRESCO (Yang et al., 2024), and TokenFlow (Geyer et al., 2023). As observed, Rerender can sometimes over-edit, resulting in strange bright spots or objects that are not in the source video. FRESCO demonstrates good temporal consistency, but it always alters the background even though the prompt doesn't mention it. This case significantly hinders the controllability of video editing. The editing results of TokenFlow, which also follows a two-step editing, are close to AdaFlow in frame quality but much inferior in temporal consistency when editing long videos. As marked by the red boxes, the editing also shows the lack of temporal consistency and defective editing quality by TokenFlow. It can be observed that the bird's beak often changes in the first editing results, indicating temporal inconsistency. In the last example, it also generates a red object that is irrelevant to the prompt and does not exist in the source video. Compared to TokenFlow and the other two baselines, our proposed AdaFlow can maintain consistency in long video editing tasks while achieving high-quality edits. Conclusively, these results show that our AdaFlow can not only achieve long video editing of more than $1k$ frames in one inference but also can obtain better video quality and consistency than existing methods.

In Fig.4, we also ablate the effect of the *Adaptive Keyframe Selection* (AKS) in AdaFlow. It can be seen that the example on the left figure shows a car quickly entering the video frame. With AKS, AdaFlow can automatically select more keyframes of this content, significantly improving image quality. The example on the right shows a constantly moving cat. Since uniform keyframe sampling is difficult to deal with such motion scenes, the cats in the generated results are always blurred. In contrast, when the cat suddenly yawns, AKS can automatically identify the rapid change and sample keyframes at this point, resulting in much better generation quality for the suddenly appearing tongue. Overall, these results confirm the effectiveness of our AdaFlow for editing videos with obvious variations.

## 6 CONCLUSION

In this paper, we present a novel and training-free method for high-quality long video editing, termed *AdaFlow*, which can effectively edit more than $1k$ video frames in one inference. By introducing the innovative designs of *Adaptive Attention Slimming* and *Adaptive Keyframe Selection*, AdaFlow significantly reduces computational resource consumption while enhancing the number of keyframes that can be edited simultaneously. We also build a new benchmark called *LongV-EVAL* to complement the evaluation of text-driven long video editing. Extensive experiments are conducted and show that AdaFlow is more effective and efficient than the compared methods in long video editing.

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

## A DATASET ANNOTATING DETAILS

We collected 75 videos, each approximately one minute long with a frame rate of 20-30 fps, from *https://mixkit.co/, https://www.pexels.com, and https://pixabay.com*. The video content spans various subjects, including people, animals, and landscapes. To annotate these data with high-quality editing prompts, we first input the video $V$ and prompt $P_1$ into Video-Llava (Lin et al., 2023), where $P_1$ is *"Please add a caption to the video in great detail."* This generates a detailed textual description $C$ of the video.

Next, we input prompt $P_2$ into GPT-4 (Achiam et al., 2023), where $P_2$ has three different forms to generate three distinct editing prompts for the same video. The forms of $P_2$ are as follows:

- *"I have a video caption: $C$. Imagine that you have modified the **main object** of the video content (such as color change, similar object replacement, etc.). After editing, add a concise one-sentence caption of the edited video (with emphasis on the edited part, no more than 15 words), not the original video content. The answer should contain only the caption, without any additional content."*

- *"I have a video caption: $C$. Imagine that you have modified the **background** of the video content (such as background tone replacement, similar background replacement, etc.). After editing, add a concise one-sentence caption of the edited video (with emphasis on the edited part, no more than 15 words), not the original video content. The answer should contain only the caption, without any additional content."*

- *"I have a video caption: $C$. Imagine that you have applied Van Gogh, Picasso, Da Vinci, Mondrian, watercolors, comics, or **drawings style transfer** to the video. After editing, add a concise one-sentence caption of the edited video (with emphasis on the style, no more than 15 words), not the original video content. The answer should contain only the caption, without any additional content."*

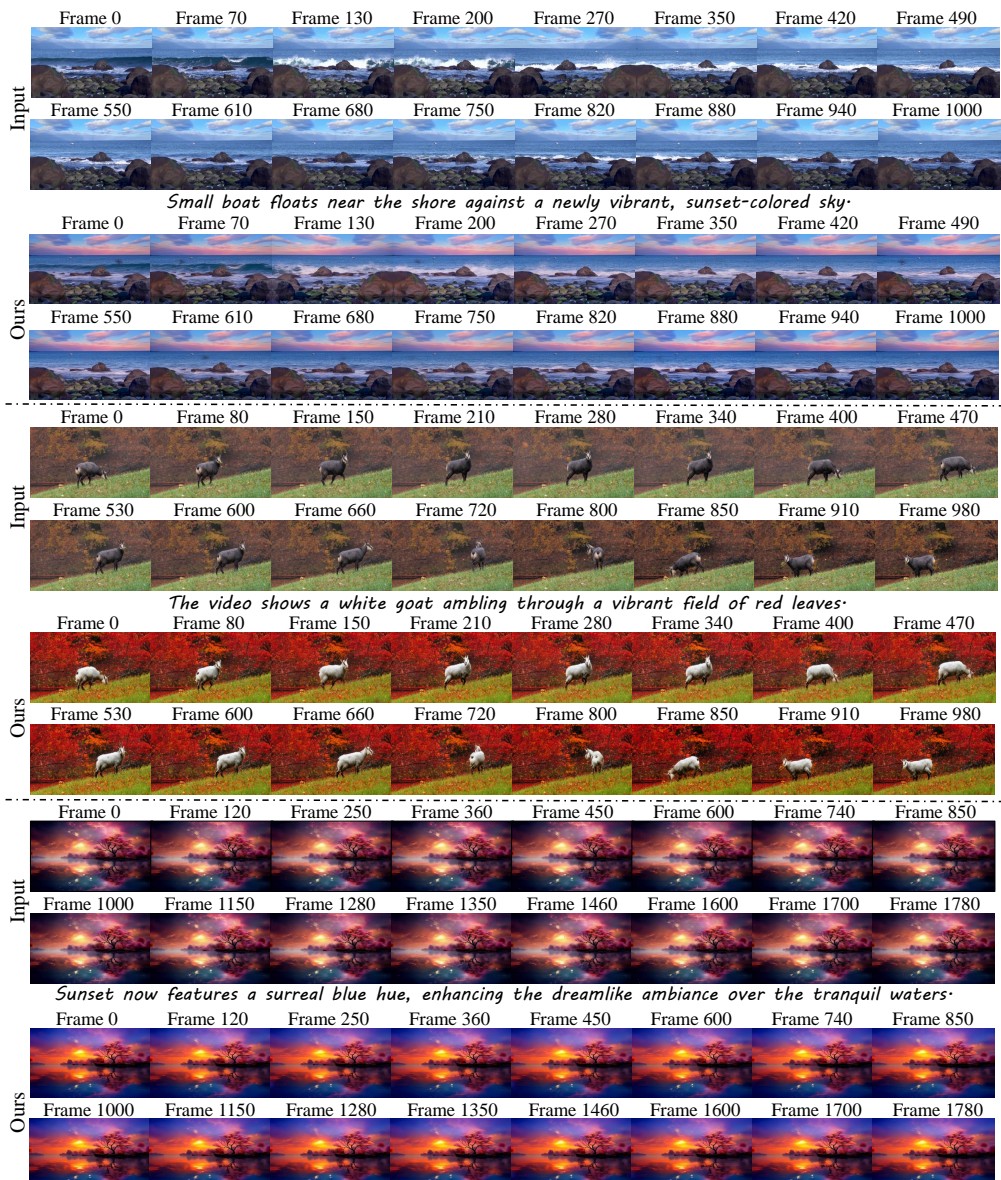

Figure 5: Additional Qualitative Results. Our method supports a wide variety of text-driven video edits and maintains high editing quality and temporal consistency even for videos exceeding a thousand frames.

This process results in three final editing prompts for each video.

## B    ADDITIONAL QUALITATIVE RESULTS

As shown in Fig.5, our method can edit over a thousand video frames on a single NVIDIA A800 (80GB) while maintaining temporal consistency and achieving high editing quality.

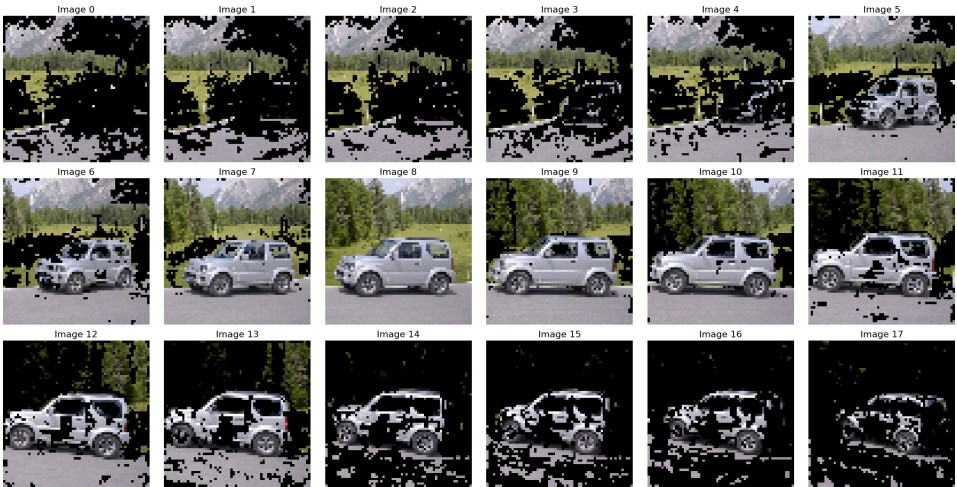

Figure 6: We retain only the tokens corresponding to the regions shown in the figure for $K$ and $V$ during the self-attention computation. In the scenario illustrated here, the eighth frame serves as the query. It can be observed that the content closer to the query frame is automatically retained more, while the content further away from the query frame is discarded more. This automatic selection can save substantial computational resources while maintaining the continuity and consistency of video generation.

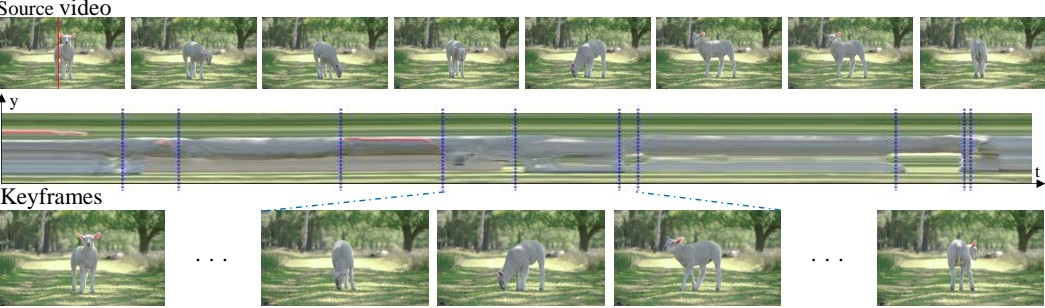

Figure 7: y-t plot. We extracted a vertical column of pixels from the center of each video frame and then sequentially stitched these columns together from left to right to get the y-t plot. The blue lines in the figure indicate the points where the video is segmented.

## C   USER STUDY DETAILS

We randomly selected 20 video-text pairs from our dataset for a user study, comparing them with the five baselines mentioned in the main text. For each pair, 50 participants were asked to evaluate and select the best video from the six options based on the following criteria:

- **Video Quality**: The edited video should appear realistic and not easily identifiable as AI-generated. Only the parts specified by the prompt should be edited, while the content not mentioned in the prompt should remain consistent with the source video.
- **Temporal Consistency**: The same object should remain consistent at any point in the long video, and the transitions between frames should be as smooth as in the source video.

## D   VISUALIZATION OF ADAPTIVE ATTENTION SLIMMING

As shown in Fig.6, the eighth frame serves as the *query* in this attention operation. By employing our proposed method, a portion of the tokens can be automatically discarded to save computational

resources. The content closer to the *query* frame is retained more, while the content further away from the *query* frame is discarded more. This is because, with a larger period, a significant amount of content dissimilar to the *query* appears in the frames, and attending to this content does not contribute to the continuity and consistency of the video. Conversely, the content closer to the query is crucial for maintaining the smoothness of the video. Therefore, using our proposed method not only saves memory but also minimally impacts the quality of video generation.

## E    VISUALIZATION OF KEYFRAME SELECTION

To visualize the *Adaptive Keyframe Selection*, we extracted a vertical column of pixels from the center of each video frame. We then sequentially stitched these columns together from left to right to create a y-t diagram, as shown in Fig.7. The blue dashed lines in the figure indicate the points where we segmented the video. It can be observed that each segmentation point corresponds to a significant change in the video content. Moreover, the keyframes obtained from each segment always contain different content. This demonstrates the effectiveness of our method.

## F    LIMITATIONS

Our method utilizes the motion information from the source video as a reference to generate non-key frames. Therefore, our approach performs exceptionally well when the image structure remains unchanged. However, it often produces unsatisfactory results when changes in object shapes are required. Additionally, since our method is training-free and directly employs image editing techniques, it primarily addresses the issue of temporal consistency. Consequently, the editing capability of our method may be influenced by the performance of the image editing techniques used.

