# OpenReview forum: "AdaFlow: Efficient Long Video Editing via Adaptive Attention Slimming And Keyframe Selection"
_ICLR.cc/2025/Conference — Submitted to ICLR 2025_

### Official Review · Reviewer_w84C · 2024-10-27

**Soundness:** 3
**Presentation:** 3
**Contribution:** 2
**Rating:** 5
**Confidence:** 4

**Summary:**

The paper proposes an efficient long video editing method capable of performing video appearance editing on 1k frames within minutes. To accomplish this, the authors developed an attention slimming + key-frame selection approach and established a long-video editing benchmark.

**Strengths:**

1. The chosen problem setting is novel, as currently there are no video-editing solutions that can handle thousands of frames effectively.

2. The efficiency optimizations are noteworthy, particularly the single correspondence computation requirement compared to other methods.

**Weaknesses:**

1. The results appear to differ from traditional video editing, resembling more appearance editing focused on texture-level modifications. Even Token-Flow, which addresses similar tasks, demonstrates more ambitious edits (e.g., woman > Pixar animation, car > ice sculpture).

2. The feature similarity-based keyframe selection doesn't appear particularly innovative.

3. Several concerns regarding the attention slimming operation:
   - The KV token selection strategy appears static regardless of the editing prompt (e.g., not specifically selecting tokens from editing-relevant regions)
   - The paper lacks comparison with other extended self-attention slimming methods, such as random sampling in StoryDiffusion
   - The correspondence-based filtering may not necessarily capture representative compressed features for the entire video (compared with random sampling)

Reference: StoryDiffusion: Consistent Self-Attention for Long-Range Image and Video Generation

**Questions:**

The paper's novelty appears limited when compared to Token-Flow, with results showing minimal differentiation except for the bike case (Fig.3, the last case). I recommend:

1. Providing 2-3 comparative examples using Token-Flow's official cases
2. Including a discussion on how different attention-slimming strategies impact the results

---

> ### Author Response · Authors · 2024-11-22
> **Response to Reviewer w84C (Part 1 of 2)**
>
> We highly appreciate your time and effort in reviewing this paper, and also thank you a lot for your constructive and encouraging comments on our work. Below, we respond to your key concerns point by point.
>
> **Q1: The results appear to differ from traditional video editing, resembling more appearance editing focused on texture-level modifications. Even Token-Flow, which addresses similar tasks, demonstrates more ambitious edits. Providing 2-3 comparative examples using Token-Flow's official cases.**
>
> **A1**: Thanks for this constructive comment. Following your suggestion, we provide more examples using TokenFlow's official cases in our anonymous project: https://anonymous.4open.science/r/AdaFlow-C28F/rebuttal/video_short.mp4.
>
> It can be seen that the editing effects of TokenFlow and AdaFlow are similar, and our AdaFlow are even better in details. For example, in the three examples given, we keep the details of a car's rearview mirror, the fingers near the basketball, and the background behind the man better. Notably, for these short videos, our adaptive attention slimming (AAS) can still be used to reduce memory overhead, and these results well confirm its effectiveness in reducing token redundancy while retaining high-quality editing.
>
> To explain, the similar effects of TokenFlow and our AdaFlow is due to the same use of PnP-Diffusion [4] as the basic image editing paradigm. But in practice, in addition to AAS and adaptive keyframe selection (AKS), our AdaFlow is also different from TokenFlow in terms of motion flow computation. As introduced in Sec. 4.3, we use the Diffusion Features (DIFT) between frames to compute the flow changes, while TokenFlow resort to all extended self-attention latents in UNet of diffusion.  This step exists in each layer of extended self-attention, and it takes about hundreds of repetitions to fully edit a video (timesteps * number of self-attention layers in a UNet). In addition, its computational complexity increases quadratically with the number of frames between adjacent keyframes. Thus, TokenFlow have to limit the keyframe interval for excessive computation, e.g., every 8 frames for one keyframe sampling. In contrast, if a certain video lacks dynamics, the keyframe interval of our method can reach hundreds of frames, which can significantly improve the computational efficiency.
>
> With these innovative designs, our advantages than TokenFlow become more obvious in editing long video. Here, we also use the official instruction to edit a long video in our anoymous project: https://anonymous.4open.science/r/AdaFlow-C28F/rebuttal/video_long.mp4.
>
> As can be seen, since TokenFlow cannot edit the whole video at once, there is a clear sense of fragmentation when editing in segments. In addition, due to the effectiveness of AAS and AKS modules in AdaFlow, our method can not only edit longer videos at once, but also the editing quality is better than TokenFlow.
>
> **Q2: The feature similarity-based keyframe selection doesn't appear particularly innovative.**
>
> **A2**: Thanks for this comment. In addition to the first attempt of adaptive keyframe selection in video editing, our AKS also greatly differs from existing video partitioning methods [1,2,3] for video understanding and retrieval in both principle and operation.
>
> In terms of principle, our AKS focuses on grasping both slight and obvious changes of video content, while the ones for video understanding often aim to truncate clips with great changes, e.g., scene changes or shot transition. For instance, we visualize some examples processed by our AKS to further confirm the above merits, which is given in our anonymous project: https://anonymous.4open.science/r/AdaFlow-C28F/rebuttal/keyframe_example2.pdf. AKS can well capture the slight changes of objects' angles in these examples, such as the turning of heads of man and cat, and even the disappearance of the tiger’s tail. This kind of small change is a concern for video editing and usually ignored in video understanding and retrieval.
>
> Meanwhile, we also differs in methodology. In addition to the different video features used, i.e., the diffusion features (DIFT) for our AdaFlow, our method fully leverages the corresponding point matching based on DIFT, instead of merely calculating feature similarity. Here, we use the confidence of the point matching as the basis.
>
> Overall, we think that our AKS is orthogonal and novel based on above discussions.

---

> ### Author Response · Authors · 2024-11-22
> **Response to Reviewer w84C (Part 2 of 2)**
>
> **Q3: The KV token selection strategy appears static regardless of the editing prompt (e.g., not specifically selecting tokens from editing-relevant regions)；The correspondence-based filtering may not necessarily capture representative compressed features for the entire video (compared with random sampling)**
>
> **A3**: Thanks for this insightful comment. In fact, our AdaFlow focuses on the token redundancy of the frame-wise extended self-attention, which is often the computation bottleneck and before prompt-based cross-attention.
>
> To explain, most of video editing methods primarily extend the self-attention module in diffusion models to all video frames, commonly referred to as *extended self-attention* to maintain the smoothness and consistency of edited videos [5,6]. Despite its effectiveness, this solution will lead to a quadratic increase in computation as the number of video frames grows, and the token-based representations of these video frames further greatly exacerbate the memory footprint. Our method is proposed to solve exactly such a problem, so it is not related to cross-attention and does not need to consider editing prompts.
>
> To better address your concern, we follow your suggestion to make a comparison with random sampling in the following table.
>
> |         | Token Pruning Ratio | Frames Quality | Video Quality |
> | :-----: | :-----------------: | :------------: | :-----------: |
> | w/o AAS |         0%          |      5.47      |     0.827     |
> | w/ AAS  |         50%         |      5.49      |     0.824     |
> | w/ AAS  |         75%         |      5.49      |     0.826     |
> | w/ RAS  |         50%         |      5.40      |     0.806     |
> | w/ RAS  |         75%         |      5.36      |     0.798     |
>
> * RAS stands for *Random Attention Slimming*..
>
> We can see that our AAS is consistently better than random sampling on both frame and video qualities. However, these quantitative results can not actually reflect their gap in video editing, so we also visualize their comparison in https://anonymous.4open.science/r/AdaFlow-C28F/rebuttal/video_ablation.mp4.
>
> It can be seen that random sampling is prone to cause degradation of editing quality and blurring of video frames. For example, in the case of random sampling, the background of the lion in the video is less edited, the person riding is very blurred, and the bird dyed blue is blurred.
>
> Overall, our AAS method can significantly save computational resources and increase the length of editable videos by several times with little loss of generation quality. The damage to generation quality is also much lower than random sampling.
>
> **Reference**
>
> [1] Wang, H., Lai, C., Sun, Y., & Ge, W. (2024, October). Weakly Supervised Gaussian Contrastive Grounding with Large Multimodal Models for Video Question Answering. In *Proceedings of the 32nd ACM International Conference on Multimedia* (pp. 5289-5298).
>
> [2] Liang, J., Meng, X., Wang, Y., Liu, C., Liu, Q., & Zhao, D. (2024). End-to-End Video Question Answering with Frame Scoring Mechanisms and Adaptive Sampling. *CoRR*.
>
> [3] Wu, M., Cao, M., Bai, Y., Zeng, Z., Chen, C., Nie, L., & Zhang, M. An Empirical Study of Frame Selection for Text-to-Video Retrieval. In *The 2023 Conference on Empirical Methods in Natural Language Processing*.
>
> [4] Narek Tumanyan, Michal Geyer, Shai Bagon, and Tali Dekel. Plug-and-play diffusion features for text-driven image-to-image translation. In *Proceedings of the IEEE/CVF Conference on Computer Vision and Pattern Recognition*, pp. 1921–1930, 2023.
>
> [5] Geyer, M., Bar-Tal, O., Bagon, S., & Dekel, T. (2023). Tokenflow: Consistent diffusion features for consistent video editing. *arXiv preprint arXiv:2307.10373*.
>
> [6] Jay Zhangjie Wu, Yixiao Ge, Xintao Wang, Stan Weixian Lei, Yuchao Gu, Yufei Shi, Wynne Hsu, Ying Shan, Xiaohu Qie, and Mike Zheng Shou. Tune-a-video: One-shot tuning of image diffusion models for text-to-video generation. In *Proceedings of the IEEE/CVF International Conference on Computer Vision*, pp. 7623–7633, 2023b.

---

> ### Author Response · Authors · 2024-11-25
> **Awaiting Your Feedback**
>
> Dear Reviewer w84C,
>
> Thanks again for your great efforts and constructive advice in reviewing this paper! With the discussion period drawing to a close, we expect your feedback and thoughts on our report. We put a significant effort into our response, with several new experiments and discussions. We sincerely hope you can consider our reply in your assessment.
>
> We look forward to hearing from you, and we can further address unclear explanations and remaining concerns if any.
>
> Regards,
>
> Authors

---

> > ### Comment · Reviewer_w84C · 2024-11-25
> >
> > Thanks for the authors' efforts for the detailed response. It already solved most of my concerns, especially the quantitative evaluation to support the novelty of AAS.
> > For the video comparison between TokenFlow and AdaFlow, I'm still worrying about the editing strength, which is lower than the TokenFlow and sometimes it also introduces some artifacts, such as the flickering at the front of the iced car, and the dog's head (which is not in a low-poly style).
> > After reading other reviewers' comments, I would like to maintain my initial evaluation of 5.

---

> > > ### Author Response · Authors · 2024-11-27
> > >
> > > Thanks a lot for your valuable feedback.
> > >
> > > From our qualitative analysis, we think that our AdaFlow is on par with TokenFlow in terms of short video editing, which is even better in some cases. As shown in our comparison with its official demos, https://anonymous.4open.science/r/AdaFlow-C28F/rebuttal/video_short.mp4, our editing about the dynamic motions of videos is more accurate, for example, the fingers of the athlete spinning the ball are clearer. Besides, over-editing is less frequent by our AdaFlow, for instance, the background details of the sculpture man are not lost. To explain, with adaptive keyframe selection (AKS), our AdaFlow can better capture important frames for translation, thereby improving better editing quality compared to the uniform-sampling based editing.
> > >
> > > In addition, our advantage is more obvious in terms of minute-long video editing. Since TokenFlow can only process about 100~200 frames in one processing, so the editing of minute-long videos needs to be separately processed in multiple runs, which are then stitched together. It can be clearly observed that there is a significant disfluency in the stitching of each video, and the temporal consistency between each video cannot be maintained. In contrast, our AdaFlow can well maintain the smoothness and consistency of long-video editing. More details can refer to https://anonymous.4open.science/r/AdaFlow-C28F/rebuttal/video_long.mp4 and https://anonymous.4open.science/api/repo/AdaFlow-C28F/file/rebuttal/video_10k.mp4?v=1044c591&download=true.
> > >
> > > In addition to the high-quality and efficient video editing achieved by our AdaFlow, we also propose a new LongV-Eval benchmark to supplement the evaluation of long video editing. In this case, we believe that the novelty and contribution of this paper are indeed significant. Moreover, our code has been anonymously given and will be publicly released after acceptance.
> > >
> > > Thanks again for your valuable time and efforts, and we look forward to your further discussions.

---

### Official Review · Reviewer_4pV2 · 2024-10-30

**Soundness:** 3
**Presentation:** 3
**Contribution:** 3
**Rating:** 6
**Confidence:** 5

**Summary:**

The proposed AdaFlow enables 1k frames editing in one inference. This is realized with adaptive keyframe selection and attention slimming. In addition, this paper developed a new benchmark, LongV-EVAL, for long video editing evaluation. Experiments proves its efficient editing performance and diversity of editing types. However, this paper avoids shape editing of the video content and only shows the effect of recoloring, which is less intensive and difficult than shape editing. Considering the importance and practical value of shape editing, it is necessary to explain whether this paper can achieve this function. If this method cannot edit shape of the objects well, then the statement in the paper about "supports foreground changes" is wrong and needs to be corrected.

**Strengths:**

1. The proposed AAS is interesting, which only focuses on the the area to be edited in KV sequences, greatly increasing GPU memory.
2. The task is meaningful and the performance is practical. By combining keyframe editing/propagation and attention slimming, this method enables 1k frames editing on a single GPU, which is impressive.

**Weaknesses:**

1. Lacking some necessary discussion. As far as I know, some neural representation based video editing can also achieves consistent long video editing (I listed some of them in Questions). These methods attempts to resolve long video editing via neural representation rather than adjusting attention in diffusion model, which also achieves impressive performance. It is necessary to discuss the advantages and disadvantages of AdaFlow and these methods.

2. Lack of necessary editing results, mainly editing the shape of objects. Although authors claimed that AdaFlow supports various editing tasks. In the paper, I only found some results of the color changing, including style transfer, background or foreground recoloring. However, various SOTA video editing methods enables deformation, that is, changing the shape of the specified objects. I believe this is a very significant function and it is much more difficult to edit than recoloring. The author should provide adequate results of editing the shape of the object (e.g. removing the foreground of a video, or turning the bird in Fig.2 into a squirrel, etc.). If this is not possible, it should be acknowledged that good deformation results cannot be achieved due to the limitations of the paper's technology.

**Questions:**

1. What are the respective advantages and disadvantages of this paper and previous neural representation-based long video editing methods in terms of performance? Some methods should be discussed [1-3].

2. Why not show the effect of shape editing on video content? This kind of editing is much stronger than the recoloring shown in the paper, and it is more difficult to achieve, but more practical.

[1] Huang J, Sigal L, Yi K M, et al. Inve: Interactive neural video editing[J]. arXiv preprint arXiv:2307.07663, 2023.

[2] Kasten Y, Ofri D, Wang O, et al. Layered neural atlases for consistent video editing[J]. ACM Transactions on Graphics (TOG), 2021, 40(6): 1-12.

[3] Yang S, Mou C, Yu J, et al. Neural video fields editing[J]. arXiv preprint arXiv:2312.08882, 2023.

---

> ### Author Response · Authors · 2024-11-22
> **Response to Reviewer 4pV2**
>
> We highly appreciate your time and effort in reviewing this paper, and also thank you a lot for your constructive and encouraging comments on our work. Below, we respond to your key concerns point by point.
>
> **Q1: Lacking some necessary discussion. As far as I know, some neural representation based video editing can also achieves consistent long video editing (I listed some of them in Questions). These methods attempts to resolve long video editing via neural representation rather than adjusting attention in diffusion model, which also achieves impressive performance. It is necessary to discuss the advantages and disadvantages of AdaFlow and these methods.**
>
> **A1**: Thanks for your constructive suggestion. We will cite and discuss your recommend excellent works in our final submission.
>
> We think the main advantage of our AdaFlow lies in efficiency. Your mentioned neural presentation-based methods requires test-time training for each video, while AdaFlow is traning-free. In practice, we run [2] for 1k frame editing, which takes about 12 hours for one video. In contrast, our AdaFlow only requires about 40 minutes.  For the other two works [1,3], they are theoretically faster than [2], but we fail to make a comparison due the lack of their codes. According to the comparison between their paradigms, we think that our AdaFlow is still much faster than these two methods.
>
> However, we agree that methods based on neural representation do have certain advantages, such as the ability of adding fine-grained elements while preserving the content of the source video. In contrast, our method is controlled by text prompts, and fine-grained details may not be perfectly described in text information, which is a limitation of the training-free methods based on diffusion models.
>
> **Q2: Lack of necessary editing results, mainly editing the shape of objects. Although authors claimed that AdaFlow supports various editing tasks. In the paper, I only found some results of the color changing, including style transfer, background or foreground recoloring. However, various SOTA video editing methods enables deformation, that is, changing the shape of the specified objects. I believe this is a very significant function and it is much more difficult to edit than recoloring. The author should provide adequate results of editing the shape of the object (e.g. removing the foreground of a video, or turning the bird in Fig.2 into a squirrel, etc.). If this is not possible, it should be acknowledged that good deformation results cannot be achieved due to the limitations of the paper's technology.**
>
> **A2**: Thanks for this insightful question. Following last question, the structural editing like the shape is indeed a challenging problem for existing training-free and training-based diffusion models [4,5,6]. Such a great modification often requires a massive amount of training data and thorough model training as attempted in some recent works [7]. Compared with these training-based approach, our AdaFlow can be capable of common editing tasks without specific tuning and long inference time, which in fact is still very cost-effective. We also states this limitation in Sec. F our appendix.
>
> To better address your concern, we also visualize more examples by our AdaFlow in the anonymous link: https://anonymous.4open.science/r/AdaFlow-C28F/rebuttal/video_short.mp4. It can be seen that our AdaFlow is capable of doing most of the obvious appearance edits, like turning cars into ice textures, men into sculptures, etc.
>
> **Reference**
>
> [1] Huang J, Sigal L, Yi K M, et al. Inve: Interactive neural video editing[J]. arXiv preprint arXiv:2307.07663, 2023.
>
> [2] Kasten Y, Ofri D, Wang O, et al. Layered neural atlases for consistent video editing[J]. ACM Transactions on Graphics (TOG), 2021, 40(6): 1-12.
>
> [3] Yang S, Mou C, Yu J, et al. Neural video fields editing[J]. arXiv preprint arXiv:2312.08882, 2023.
>
> [4] Yang, S., Zhou, Y., Liu, Z., & Loy, C. C. (2024). FRESCO: Spatial-Temporal Correspondence for Zero-Shot Video Translation. In *Proceedings of the IEEE/CVF Conference on Computer Vision and Pattern Recognition* (pp. 8703-8712).
>
> [5] Cong, Y., Xu, M., Chen, S., Ren, J., Xie, Y., Perez-Rua, J. M., ... & He, S. FLATTEN: optical FLow-guided ATTENtion for consistent text-to-video editing. In *The Twelfth International Conference on Learning Representations*.
>
> [6] Geyer, M., Bar-Tal, O., Bagon, S., & Dekel, T. (2023). Tokenflow: Consistent diffusion features for consistent video editing. *arXiv preprint arXiv:2307.10373*.
>
> [7] Gu, Y., Zhou, Y., Wu, B., Yu, L., Liu, J. W., Zhao, R., ... & Tang, K. (2024). Videoswap: Customized video subject swapping with interactive semantic point correspondence. In *Proceedings of the IEEE/CVF Conference on Computer Vision and Pattern Recognition* (pp. 7621-7630).

---

> > ### Comment · Reviewer_4pV2 · 2024-11-25
> >
> > After watching your provided videos, I am confident that your method can only achieve a level of appearance editing that involves adjusting colors and textures. As you admit, modifying the structure is challenging, and Adaflow cannot do it. I believe this function is much more important than changing texture or color but unfortunately, Adaflow is unable to do so. Considering its limited application, I believe that AdaFlow's claim in the paper that it can support multiple edits is inappropriate and raises suspicion of exaggerating its own contributions. I strongly recommend the author to acknowledge in a prominent position in the main paper (including the introduction, methods, etc.) that the method cannot achieve structural editing due to the limitations of XX technology. BTW, in A2, you said **Such a great modification often requires a massive amount of training data and thorough model training**, which is also inaccurate, some video-editing methods (like FateZero) can achieve structural editing without massive and thorough training. I understand that this exaggerated description is meant to demonstrate that your method is outstanding in the direction of training-free video editing, in order to facilitate the acceptance of your paper. However, your inaccurate description harms other outstanding works and should be avoided.
> >
> > In short, although your method cannot achieve structural editing, it performs well in color, style, and texture editing. As a training-free method (or a TokenFlow-extended method), I am willing to give a score slightly higher than the acceptance threshold. I hope the authors can acknowledge the limitations of your method in a prominent position in the main paper, which is responsible to the entire AIGC community. Meanwhile, I believe that these shortcomings will not affect the quality of your method itself, it is an interesting paper :)

---

> > > ### Comment · Reviewer_4pV2 · 2024-11-25
> > >
> > > BTW, in the training-free video editing method, AdaFlow's application scope is also weaker than FateZero (ICCV 2023), which can achieve various edits including structural editing. So I think it is inappropriate for AdaFlow to roughly state in its paper that it supports multiple editors, and its limitations should be acknowledged.
> > >
> > > Apart from its limitation, I still acknowledge the novelty and contribution of AdaFlow, which is interesting and relatively novel, but not important enough. So I maintain my score as 6.

---

> > > > ### Author Response · Authors · 2024-11-26
> > > >
> > > > Thanks for your kind and constructive suggestions. We have revised our paper in Line 160-161, 191-195 and 353-355 to state the limitation of our AdaFlow in great video content modification.
> > > >
> > > > In particular, we attribute this shortcoming to the principle of motion-flow-based video editing, which extracts flow information from the source video, making it hard to make great changes to video semantic structures.
> > > >
> > > > However, the merit of motion-flow-based editing is that it can easily achieve excellent temporal consistency and inter-frame coherence based on the image generation or editing model with only the motion-flow information. Not only are these methods often training-free, they also require no additional test-time tuning with high expenditure.

---

### Official Review · Reviewer_EZqB · 2024-10-31

**Soundness:** 3
**Presentation:** 2
**Contribution:** 2
**Rating:** 5
**Confidence:** 5

**Summary:**

This paper introduces AdaFlow, a training-free method designed to address memory constraints in long video editing. AdaFlow incorporates two key strategies: Adaptive Attention Slimming, which selectively reduces token use in self-attention to decrease memory requirements, and Adaptive Keyframe Selection, which optimizes frame selection for enhanced editing quality. According to the authors, these techniques enable AdaFlow to handle videos exceeding 1,000 frames on a single A800 GPU, reportedly achieving lengths ten times greater than prior methods. The paper also presents LongV-EVAL, a benchmark for assessing long video edits, where AdaFlow demonstrates potential advantages in both efficiency and quality over existing approaches.

**Strengths:**

- The paper presents AdaFlow, a training-free approach that effectively addresses memory constraints, offering a feasible alternative for handling longer videos compared to traditional methods.

- AdaFlow combines Adaptive Attention Slimming and Adaptive Keyframe Selection to optimize memory usage and frame selection, respectively. This combined approach not only reduces computational load by focusing on essential tokens in self-attention but also enhances editing quality by selecting keyframes that capture critical scene changes.

- The introduction of LongV-EVAL provides the field with a dedicated benchmark for long video editing, complete with detailed annotations and varied scenarios, which can serve as a valuable tool for assessing future developments in this area.

- Initial results on LongV-EVAL indicate that AdaFlow may outperform existing methods in both efficiency and quality, positioning it as a promising approach for text-driven long video editing.

**Weaknesses:**

1. **Insufficient Evidence for Claimed Contributions:** Although the paper lists three main contributions, some are not substantiated in sufficient detail. For instance, the claim of “effective” memory optimization mainly references spatial memory savings, with no discussion of runtime performance or the extra computation that Adaptive Keyframe Selection  and Adaptive Attention Slimming might add. Additionally, the benchmark’s description in the paper is limited, with definitions for key evaluation metrics lacking clarity.

2. **Limited and Incomplete Experiment Comparisons:** The experiments primarily compare AdaFlow to methods focused on consistency in short video editing, lacking comparisons to dedicated long video editing techniques. Furthermore, the ablation study is minimal, with few quantitative measures and no in-depth analysis of other components within the approach. Relying on visual clarity in a few cases does not provide sufficient evidence of AdaFlow’s overall performance.

3. **Unclear Visualization and Algorithm Details:** The pipeline visualization, particularly for the AKS module, is difficult to interpret, and specific steps like the "window_check" are not well-explained, leaving some ambiguity regarding the AKS process and its impact on overall results.

4. **Limitations of Feature-Matched Latent Propagation:** If the intended edits involve adding new content or altering major background elements, rather than just subtle changes (e.g., color or style shifts), the proposed feature-matched propagation approach may fail to preserve coherence, potentially limiting its applicability for more complex or structural video modifications.

**Questions:**

See weakness.

---

> ### Author Response · Authors · 2024-11-22
> **Response to Reviewer EZqB (Part 1 of 2)**
>
> Thanks for your time and effort in reviewing this paper, and we respond to your key concern point by point below.
>
> **Q1: Insufficient Evidence for Claimed Contributions: Although the paper lists three main contributions, some are not substantiated in sufficient detail. For instance, the claim of “effective” memory optimization mainly references spatial memory savings, with no discussion of runtime performance or the extra computation that Adaptive Keyframe Selection and Adaptive Attention Slimming might add. Additionally, the benchmark’s description in the paper is limited, with definitions for key evaluation metrics lacking clarity.**
>
> **A1**: Thanks for your comment. In fact, our claims and contributions are fully supported in this paper.
>
> In terms of effective memory optimization, the effective editing of more than *1k* frames in one GPU inference already confirms the efficiency and effectiveness of our method AdaFlow and two key designs, i.e., Adaptive Keyframe Selection and Adaptive Attention Slimming. To our best knowledge, this memory efficiency is achieved for the first time.
>
> Besides, the runtime comparison between our AdaFlow and the compared methods have been given in Tab.1. Our AdaFlow is about 2-3 times faster than the compared methods.
>
> In terms of the benchmark and metrics used, we spent about two paragraphes for their introductions in Sec. 5.1. Moreover, these metrics are the widely-used ones in video editing and generation.
>
> **Q2: Limited and Incomplete Experiment Comparisons: The experiments primarily compare AdaFlow to methods focused on consistency in short video editing, lacking comparisons to dedicated long video editing techniques. Furthermore, the ablation study is minimal, with few quantitative measures and no in-depth analysis of other components within the approach. Relying on visual clarity in a few cases does not provide sufficient evidence of AdaFlow’s overall performance.**
>
> **A2**: Thanks for this comment. In this paper, we compare most SOTA text-driven video editing methods, e.g., FRESCO[1] and Flatten[2]. We are also pleased to compare more text-driven video editing methods that can handle minute-long videos. Your recommendations are welcome.
>
> In terms of ablation, we qualitatively compare the effect of Adaptive Keyframe Selection in Fig.4, showing its benefits for improving editing quality and consistency. To better answer your concern, we also report the ablation of Adaptive Keyframe Selection for 128-frame editing in the following table.
>
> |         | Token Pruning Ratio | Frames Quality | Object Consistency | Video Quality | Semantic Consistency |
> | :-----: | :-----------------: | :------------: | :----------------: | :-----------: | :------------------: |
> | w/o AAS |         0%          |      5.47      |       0.957        |     0.827     |        0.971         |
> | w/ AAS  |         50%         |      5.49      |       0.952        |     0.824     |        0.969         |
> | w/ AAS  |         75%         |      5.49      |       0.950        |     0.826     |        0.968         |
>
> It can be seen that our AAS can retain almost the same editing quality with up to 75% token pruning, which also well confirm our motivation about the token redundancy in keyframe translation.
>
> Following your suggestion, we will supplement these results to our final submission.

---

> ### Author Response · Authors · 2024-11-22
> **Response to Reviewer EZqB (Part 2 of 2)**
>
> **Q3: Unclear Visualization and Algorithm Details: The pipeline visualization, particularly for the AKS module, is difficult to interpret, and specific steps like the "window_check" are not well-explained, leaving some ambiguity regarding the AKS process and its impact on overall results.**
>
> **A3**: Thanks for this comment. AKS is to select the representative keyframes of a given video based on the diffusion feature, of which principle and process are also well described in Sec.4.1. We will follow your suggestion to revise Fig. 2 and the algorithm for clearer presentation.
>
> **Q4: Limitations of Feature-Matched Latent Propagation: If the intended edits involve adding new content or altering major background elements, rather than just subtle changes (e.g., color or style shifts), the proposed feature-matched propagation approach may fail to preserve coherence, potentially limiting its applicability for more complex or structural video modifications.**
>
> **A4**: Thanks for your comment. As discussed in the Sec.F of Appendix F, your mentioned limitation is an open problem in training-free based video editing research [2,3]. As a train-free method, AdaFlow can already achieve great modifications of a given video according to the prompt. To better address your concern, we give more editing examples compared with the SOTA method TokenFlow in our anonymous code project: https://anonymous.4open.science/r/AdaFlow-C28F/rebuttal/video_short.mp4.
>
> **Reference**
>
> [1] Yang, S., Zhou, Y., Liu, Z., & Loy, C. C. (2024). FRESCO: Spatial-Temporal Correspondence for Zero-Shot Video Translation. In *Proceedings of the IEEE/CVF Conference on Computer Vision and Pattern Recognition* (pp. 8703-8712).
>
> [2] Cong, Y., Xu, M., Chen, S., Ren, J., Xie, Y., Perez-Rua, J. M., ... & He, S. FLATTEN: optical FLow-guided ATTENtion for consistent text-to-video editing. In *The Twelfth International Conference on Learning Representations*.
>
> [3] Geyer, M., Bar-Tal, O., Bagon, S., & Dekel, T. (2023). Tokenflow: Consistent diffusion features for consistent video editing. *arXiv preprint arXiv:2307.10373*.

---

> ### Author Response · Authors · 2024-11-25
> **Awaiting Your Feedback**
>
> Dear Reviewer EZqB,
>
> Thanks again for your great efforts and constructive advice in reviewing this paper! With the discussion period drawing to a close, we expect your feedback and thoughts on our report. We put a significant effort into our response, with several new experiments and discussions. We sincerely hope you can consider our reply in your assessment.
>
> We look forward to hearing from you, and we can further address unclear explanations and remaining concerns if any.
>
> Regards,
>
> Authors

---

> > ### Comment · Reviewer_EZqB · 2024-11-25
> > **Thank you for your response**
> >
> > I appreciate your detailed reply. It has addressed most of my concerns, and I will raise my score accordingly.

---

> ### Author Response · Authors · 2024-11-26
> **Response to Reviewer EZqB**
>
> Thanks a lot for your response, and your time and efforts are both highly appreciated.
>
> We would like to emphasize our contributions again: 1) We propose a novel video editing approach called AdaFlow with two innovative designs namely Adaptive Attention Slimming (AAS) and Adaptive Key-frame Selection, which achieves high-quality video editing of more than 1k frames for the first time. 2) A new benchmark called LongV-Eval is also proposed to supplement the existing evaluation systems of video editing.
>
> Our work is also highly recognized by other reviewers, such as "The proposed adaptive keyframe selection based on dynamic content is a valuable contribution'' and "The proposed Long-V-Eval benchmark is valuable for advancing future video editing research'' by R#BVZX; "The proposed AAS is interesting, which only focuses on the area to be edited in KV sequences, greatly increasing GPU memory ", "The task is meaningful and the performance is practical" and "this method enables 1k frames editing on a single GPU, which is impressive." by R#4pV2; "The chosen problem setting is novel" and "The efficiency optimizations are noteworthy" by R#w84C.
>
> We look forward to your further discussions and answering your other concerns.
>
> Regards
>
> The authors

---

> ### Comment · Reviewer_EZqB · 2024-12-02
> **Thank you for your response**
>
> Most of the issues (2, 3, 4) have been addressed, which justifies my increase to 5. However, I still have concerns regarding the efficiency comparison of AKS. Table 1 only reports overall runtime, and some compared methods are not specifically optimized for long video editing. Applying them directly to 1,000-frame editing without targeted adjustments weakens the comparison’s persuasiveness.
>
> For instance, it would be more convincing to include quantitative ablation analyses on the overhead and effectiveness of AKS and AAS, similar to the Table 3 in "Rerender A Video" [1]. Additionally, aligning the number of keyframes selected in AKS with other methods' frame selection strategies for a direct comparison would strengthen the argument.
>
> Based on these reasons, I will maintain my current score.
>
> [1] Yang, S., Zhou, Y., Liu, Z., & Loy, C. C. (2023). Rerender A Video: Zero-Shot Text-Guided Video-to-Video Translation. ACM SIGGRAPH Asia Conference Proceedings.

---

> > ### Author Response · Authors · 2024-12-03
> >
> > Thanks a lot for your valuable feedback.
> >
> > In Table 1, we report the average time overhead of different video editing methods on the proposed LongV-EVAL, which includes all processing steps, e.g., AKS and AAS in our AdaFlow. It can be seen that our method is about 2-4 times faster than the compared methods, including your mentioned Rerender.
> >
> > To better respond to your concerns, we report the detailed running time of our AdaFlow's designs on one short and one long video demos in the following tables, and TokenFlow is used as reference.
> >
> > **Short Video** (About 120 frames, https://anonymous.4open.science/r/AdaFlow-C28F/rebuttal/video_short.mp4)
> >
> > |           | Inversion | AKS  | AAS  | Diffusion Editing |  All   |
> > | :-------: | :-------: | :--: | :--: | :---------------: | :----: |
> > |  AdaFlow  |   1min    | 4sec | 5sec |      3.6min       | 4.8min |
> > | TokenFlow |   1min    |  -   |  -   |      5.9min       | 6.9min |
> >
> > **Long Video** (About 1000 frames, https://anonymous.4open.science/r/AdaFlow-C28F/rebuttal/video_long.mp4)
> >
> > |           | Inversion |  AKS  | AAS  | Diffusion Editing |  All  |
> > | :-------: | :-------: | :---: | :--: | :---------------: | :---: |
> > |  AdaFlow  |   7min    | 33sec | 1min |       16min       | 24min |
> > | TokenFlow |   7min    |   -   |  -   |       33min       | 40min |
> >
> > It can be seen that AKS and AAS both take much less time than the diffusion editing process. Moreover, they can help to reduce the time of diffusion editing, as discussed in the paper.
> >
> > In our previous discussions, i.e. Q2, we have supplemented the quantitative ablation results of AAS.
> >
> > |         | Token Pruning Ratio | Frames Quality | Object Consistency | Video Quality | Semantic Consistency |
> > | :-----: | :-----------------: | :------------: | :----------------: | :-----------: | :------------------: |
> > | w/o AAS |         0%          |      5.47      |       0.957        |     0.827     |        0.971         |
> > | w/ AAS  |         50%         |      5.49      |       0.952        |     0.824     |        0.969         |
> > | w/ AAS  |         75%         |      5.49      |       0.950        |     0.826     |        0.968         |
> >
> > We can find that with a high pruning ratio, AAS can well maintain the editing quality and greatly reduce the computation and memory overhead. Besides, the effects of our AKS can be found in Fig. 4, which can better capture the subtle changes of video content for better editing quality.
> >
> > Overall, we believe that these results can well confirm the efficiency and effectiveness of our designs. Following your suggestion, we will supplement these results to our final submission.

---

### Official Review · Reviewer_BVZX · 2024-11-02

**Soundness:** 3
**Presentation:** 3
**Contribution:** 3
**Rating:** 6
**Confidence:** 4

**Summary:**

This paper addresses the text-based video editing problem from a new angle compared to prior work, emphasizing keyframe selection and efficiency for generating longer videos. Building on keyframe translation and interpolation approaches from previous studies, the authors propose an Adaptive Keyframe Selection and Adaptive Attention Slimming scheme to enhance quality and handle longer videos with more transitions. The proposed method achieves superior results and efficiency compared to TokenFlow and other baselines. Additionally, the authors introduce a benchmark, LongV-EVAL, for evaluating long video editing tasks.

**Strengths:**

- Keyframe selection is an important topic in video editing and remains underexplored in prior work. The proposed adaptive keyframe selection based on dynamic content is a valuable contribution.
- The proposed Long-V-Eval benchmark is valuable for advancing future video editing research.
- The evaluation results in Table 1 and Table 2 outperform prior methods. Additionally, the supplementary videos demonstrate the proposed method's ability to achieve high visual quality.

**Weaknesses:**

- Additional examples of keyframe selection would clarify the approach, such as cases where changes in object angle or the appearance of new objects lead to their selection as keyframes.
- The results appear quite similar to TokenFlow; further clarification on this similarity would be beneficial. Also, the videos in the supplementary materials look slightly oversmoothed, with some detail loss and minor color mixing from the background compared to TokenFlow.
- The ablation study is somewhat limited. There is no analysis of adaptive attention slimming, and it would be beneficial to include quantitative results from the ablation study.

**Questions:**

- What is the maximum number of frames each baseline can handle?
- For the longer videos in the supplementary materials, are the baseline videos created by simply concatenating shorter clips?

---

> ### Author Response · Authors · 2024-11-22
> **Response to Reviewer BVZX (Part 1 of 2)**
>
> We highly appreciate your time and effort in reviewing this paper, and also thank you a lot for your constructive and encouraging comments on our work. Below, we respond to your key concerns point by point.
>
> **Q1: Additional examples of keyframe selection would clarify the approach, such as cases where changes in object angle or the appearance of new objects lead to their selection as keyframes.**
>
> **A1**: Thanks for your constructive suggestion. The capture of sudden changes in object angle or the appearance of new objects is indeed the main advantage of our *adaptive keyframe selection* (AKS) method. Following your suggestion, we visualize two groups of examples processed by AKS to further confirm this advantage, which is given in our anonymous project: https://anonymous.4open.science/r/AdaFlow-C28F/rebuttal/keyframe_example1.pdf.
>
> In the first group of examples, we show that AKS can accurately capture the appearance of new objects with its DIFT-based algorithm proposed in the paper, which is visualized in the last column. Similarly, the second group also shows that our AKS can well capture the subtle changes of objects' angles in the second group, such as the head turning of the man and the cat, and even the disappearance of the tiger’s tail.
>
> Overall, these visualizations confirm again the merit of our AKS in key frame selection, which will be supplemented to our Appendix of our final submission.
>
> **Q2: The results appear quite similar to TokenFlow; further clarification on this similarity would be beneficial. Also, the videos in the supplementary materials look slightly oversmoothed, with some detail loss and minor color mixing from the background compared to TokenFlow.**
>
> **A2**: Thanks for this comment. The main reason why the editing results are similar is that both TokenFlow and our AdaFlow are based on PnP-Diffusion [1] for image editing, while the other compared methods directly apply Stable Diffusion instead of image editing methods.
>
> In addition to AKS, our AdaFlow is also different from TokenFlow in terms of motion flow computation. As introduced in Sec. 4.3, we use the *Diffusion Features* (DIFT) between frames to compute the flow changes, while TokenFlow resort to all *extended self-attention* latents in UNet. This step exists in each layer of extended self-attention, and it takes about hundreds of repetitions to edit a video (timesteps * number of self-attention layers in a UNet). In addition, its computational complexity increases quadratically with the number of frames between adjacent keyframes. Thus, TokenFlow have to limit the keyframe interval for excessive computation, e.g., every 8 frames for one keyframe sampling. In contrast,  if a certain video lacks dynamics, the keyframe interval of our method can reach hundreds of frames, which can significantly improve the computational efficiency.
>
> Meanwhile, through our empirical observations, the editing quality of two methods are in fact close, and in some cases ours are better, which is given in our anonymous project: https://anonymous.4open.science/r/AdaFlow-C28F/rebuttal/video_short.mp4. Moreover, our AdaFlow also achieve the high-quality editing of minute-long videos for the first time.
>
> **Q3: The ablation study is somewhat limited. There is no analysis of adaptive attention slimming, and it would be beneficial to include quantitative results from the ablation study.**
>
> **A3**: Thanks for this suggestion. In particular, without adaptive attention slimming (AAS), AdaFlow is incapable of long video editing. To address your concern, we report its ablation study on the top-128-frame editing of all videos. The ablation results are given below.
>
> |         | Token Pruning Ratio | Frames Quality | Object Consistency | Video Quality | Semantic Consistency |
> | :-----: | :-----------------: | :------------: | :----------------: | :-----------: | :------------------: |
> | w/o AAS |         0%          |      5.47      |       0.957        |     0.827     |        0.971         |
> | w/ AAS  |         50%         |      5.49      |       0.952        |     0.824     |        0.969         |
> | w/ AAS  |         75%         |      5.49      |       0.950        |     0.826     |        0.968         |
>
> It can be seen that our AAS can retain almost the same editing quality with up to 75% token pruning, which also well confirm our motivation about the token redundancy in keyframe translation.
>
> Following your suggestion, we will supplement these results to our final submission.

---

> ### Author Response · Authors · 2024-11-22
> **Response to Reviewer BVZX (Part 2 of 2)**
>
> **Q4: What is the maximum number of frames each baseline can handle?**
>
> **A4**: Thanks for your comment. For Rerender and FRESCO, their code projects can also support the editing of LongV-EVAL, i.e., up to 3k frames. However, we notice that they can only support 32 frames for diffusion editing on one GPU, while the rest frames are sequentially interpolated via Ebsynth [2] on CPU. So from Tab.1, we can see their processes are very time consuming. For TokenFlow, FLATTEN, and RAVE, the maximum number of edited frames for one inference is about 128, 32,16, respectively.
>
> In stark contrast, our AdaFlow can process thousands of frames in one inference, which is up to *10k* frames, as shown in our anonymous project: https://anonymous.4open.science/api/repo/AdaFlow-C28F/file/rebuttal/video_10k.mp4?v=1044c591&download=true.
>
> **Q5: For the longer videos in the supplementary materials, are the baseline videos created by simply concatenating shorter clips?**
>
> **A5**: Thanks for this comment. Following Q4, the results generated by Rerender and FRESCO are not stitched, while the ones of the other three methods are stitched.
>
> **Reference**
>
> [1] Narek Tumanyan, Michal Geyer, Shai Bagon, and Tali Dekel. Plug-and-play diffusion features for text-driven image-to-image translation. In *Proceedings of the IEEE/CVF Conference on Computer Vision and Pattern Recognition*, pp. 1921–1930, 2023.
>
> [2] Jamriška, O., Sochorová, Š., Texler, O., Lukáč, M., Fišer, J., Lu, J., ... & Sýkora, D. (2019). Stylizing video by example. *ACM Transactions on Graphics (TOG)*, *38*(4), 1-11.

---

> > ### Comment · Reviewer_BVZX · 2024-11-27
> >
> > Thank you to the authors for the detailed response. I agree with Reviewer 4Vp2 that while the contribution of AdaFlow is novel and interesting, its significance is limited. I would lean towards a borderline or weak accept, with a slightly positive inclination.

---

### Meta-Review · Area_Chair_r3oc · 2024-12-21

**Metareview:**

The paper aims to improve long video editing by introducing: 1) Adaptive Keyframe Selection, which replaces random selection in prior works, and 2) Adaptive Attention Slimming, which reduces the tokens in self-attention, thereby decreasing memory overhead. Additionally, the authors introduce a new benchmark dataset for long video editing.

Strength
* Keyframe selection is an important problem in various video-related tasks, and the proposed keyframe selection method is potentially useful for a range of problems.
* The proposed attention slimming method is interesting and proven effective in improving efficiency.
* The new long video editing benchmark is valuable for future research.

Weakness
* The keyframe selection results presented in the paper are limited, raising questions about the robustness and generalizability of the approach.
* The approach is similar to TokenFlow but has worse qualitative results.
* The paper lacks 1) comprehensive discussion and comparison with relevant state-of-the-art methods, and 2) ablation study
* The reviewers raise concerns about the method’s ability to handle a broad spectrum of editing tasks. The rebuttal did not resolve the concern, and the applicability of the proposed method remain questionable.

While the reviewers acknowledge the contributions of adaptive keyframe selection and adaptive attention slimming, they remain concerned about the editing capabilities of the proposed method. The limited range of edits demonstrated, as well as the method’s similarity to TokenFlow, were not fully addressed during the rebuttal. As a result, the reviewers generally agree that the impact and significance of this work appear limited.

**Additional Comments On Reviewer Discussion:**

* The authors provided additional examples for keyframe selection, partially alleviating concerns about the approach’s generalizability and robustness.
* The authors clarified the differences between the proposed method and TokenFlow; however, questions remain about the overall significance of the contributions.
* Additional discussion comparing relevant methods was provided, emphasizing the efficiency of the proposed method.
* The authors argued that certain editing tasks are difficult for current video editing methods. The authors also provide additional editing examples. However, the reviewers remain unconvinced about the method’s editing capabilities and consider the claim of supporting multiple types of editing to be overclaimed.

---

### Decision · Program_Chairs · 2025-01-22

Reject